# A Three-Dimensional Evaluation Model of the Externalities of Highway Infrastructures to Capture the Temporal and Spatial Distance to Optimal—A Case Study of China

Lei Zhu [1], Lina Zhang [1], Qianwen Ye [2], Jing Du [1] and Xianbo Zhao [3,*]

[1] Department of Construction and Real Estate, Southeast University, 2 Sipailou, Nanjing 210096, China; seuzhulei@seu.edu.cn (L.Z.); 220191235@seu.edu.cn (L.Z.); cathy_djnj@163.com (J.D.)
[2] Engineering Audit Division, China Pharmaceutical University, 639 Longmian Avenue, Jiangning District, Nanjing 211121, China; 15850617398@163.com
[3] School of Engineering and Technology, Central Queensland University, Sydney, NSW 2000, Australia
[*] Correspondence: b.zhao@cqu.edu.au

**Abstract:** Various externalities caused by highway infrastructures, such as promoting economic development, traffic congestion, and air pollution, are becoming more and more important. Currently, there is no multi-dimensional quantitative evaluation of the externalities of highway infrastructures, hindering the sustainable planning and development of highway infrastructures. Therefore, this study aims to develop a three-dimensional evaluation model of the externalities of highway infrastructures. To achieve the above objective, this study: (1) developed a three-dimensional evaluation index system through a comprehensive literature review and interviews with experts; (2) weighted the evaluation indexes using the entropy weight method; (3) developed the comprehensive evaluation model using the grey correlation analysis method; (4) validated the developed model by using statistical data of Jiangsu province, China. The analysis results showed that the developed model is feasible and effective in evaluating the externalities of highway infrastructures as the analysis results are consistent with reality. In addition, the model can capture the value of externality-related information, the distance to the optimal state of the externalities of highway infrastructures, and the temporal and spatial trends of the externalities of highway infrastructures for a region. The results of this study for the first time set a basis for investigating the influential mechanism of the multi-dimensional externalities of highway infrastructures. Moreover, the results provide theoretical support for the scientific formulation of relevant policies and decision-making for the government.

**Keywords:** highway infrastructures; externality; evaluation; entropy weight; grey correlation

## 1. Introduction

Highway infrastructures are very crude instruments of economic development and social changes. During the rapid development, highway infrastructures also produce varying degrees of external effects to the social development [1,2] and ecological environment [3–6]. For example, the promotion of the economic development of area region and the traffic convenience of the public are the positive external effects; traffic congestion, noise, and air pollution are highly concerned negative external effects. These external effects are usually defined as externalities. Some costs related to highway infrastructures are fully internalized by economic agents, such as construction costs and operation costs. Externality costs of highway infrastructures are not fully internalized and reflected by the economic transaction. However, their influence is real and cannot be ignored.

To cope with these externalities, relevant authorities have taken many response measures. For instance, building new roads, widening the existing highways, and traffic diversion are measures for improving traffic congestion and accidents. Considering the unbalanced economic development, the transportation authority vigorously invests in

the construction of highway infrastructure for the undeveloped areas to leverage the advantages of convenient transportation on economic development [7]. In terms of environmental pollution and ecological damage, policies for highway toll [4], motor vehicle emission limits [8], schemes for the shelter forest along the highway [9], and the financing model considering wetlands costs to mitigate highway runoff pollution [10] have been formulated. However, the practical results demonstrate that the effect of the governance or internalization of these externalities is not ideal. One of the possible reasons lies in the lack of a comprehensive quantitative evaluation of economic, social, and ecological externalities. Without such comprehensive evaluation, the influencing factors and the formation mechanism of the externalities of highway infrastructures cannot be accurately quantified, which is not conducive to the governance or internalization of external effects and the formulation of countermeasures with long-term and sustainable effects [11].

With the deepening of relevant research and the advance in information technology, more and more scholars and policymakers take the externalities of highway infrastructures into their traffic costs function, bringing external or social costs to internal or private costs [12,13]. However, among the economic, social, and ecological dimensions of externalities of highway infrastructures, current research mainly focuses on one or two dimensions. There has been no three-dimensional quantitative evaluation of the externalities of highway infrastructures [2,5,7,14–17], which hinders the need and possibilities for broadening the scope of highway planning by considering the three-dimensional externalities of highway infrastructures.

To extend the literature and knowledge body, this study, therefore, aims to develop a three-dimensional evaluation model of the externalities of highway infrastructures. The results will first pave the way for the research on the influencing factors and the formation mechanism of the three-dimensional externalities of highway infrastructures. Moreover, the results will provide theoretical support for the planner to make targeted and sustainable planning and for the authorities to formulate scientific policies. To achieve the above aim, the objectives of this study include: (1) developing a three-dimensional evaluation index system through a comprehensive literature review and interviews with experts; (2) weighting the evaluation indexes using the entropy weight method; (3) developing the comprehensive evaluation model using the grey correlation analysis method; (4) validating the developed evaluation model by using the relevant statistical data from Jiangsu province, China.

## 2. Literature Review

This study conducted a comprehensive literature review of the evaluation of the externalities of highway infrastructures. This study used the keywords of "externalit*" and "highway" to perform an article search under all fields using Web of Science (WoS) without time limitation. The use of an asterisk (*) in a word means that any character group can be represented. After reading the abstract of the searched articles, this study finally selected 54 articles that are most relevant to the externality evaluation of highway infrastructures. A summary of the status of the evaluation of the externalities of highway infrastructures is from four perspectives: social evaluation, economic evaluation, ecological evaluation, and comprehensive evaluation. This literature review and summary also contribute to obtaining the proper evaluation index and suitable methods.

### 2.1. Evaluation of Social Externalities

From the social aspect, previous studies have shown that highway infrastructures have external effects on social development and sustainability [1,16]. Social sustainability relates to personal characters, which may include employment and income, education, health care, skills, communication, and recreations [16]. From the perspective of welfare economics, Verhoef [18] estimated the external effects and social costs of road transport including congestion, accidents, noise, and air pollution. Using the multiple regression methods, Percoco [7] conducted an empirical study to explore the variation in employment,

population, and plants induced by the construction of the highway network by using relevant data from 1951 to 2001. The results indicated that access to a highway for a city has a positive impact on urban development, in terms of employment growth (+4–5%) and firm entry (+2–3%). Abdel-Raheem and Ramsbottom [1] analyzed the performance of highway projects concerning the social dimension of sustainability. Quality of living, diversity with employees, and awareness of social sustainability were identified as three principal contributors of highway infrastructures to social sustainability. Rostamnezhad et al. [16] proposed a new hybrid system dynamic (SD)-fuzzy decision-making trial and evaluation laboratory (DEMATEL) method to analyze the various factors affecting the social sustainability of the highway construction project. Considering the limitations in sample size, the hypotheses of variable independence in statistical methods, and the vagueness and subjectivity of experts' opinions and expressions, system dynamic and fuzzy method rather than structural equation modeling were adopted to assess the complex interactions among critical factors.

### 2.2. Evaluation of Economic Externalities

From the economic perspective, previous studies have indicated that externalities of road transport have a significant impact on economic development. Transportation investments affect the economy through increasing accessibility, mobility, safety, and travel reliability [15]. Economic impacts of transportation investments are often studied using the theories of spatial economics and the theoretical framework of production function [15]. Many empirical studies estimate the strong positive influence of transportation investments on economic output at the national, state, or county level by using elasticity estimates. The values of the elasticity range from 0.2 to 0.58, indicating a unit increase in transportation investments is associated with a 0.20 to 0.58 unit increase in economic outputs [15,19,20]. In terms of highway infrastructures, Duranton et al. [21] proved that highways within cities have a large effect on the weight of city exports with an elasticity of approximately 0.5, indicating a unit increase in highway investments is associated with a 0.5 unit increase in economic outputs.

From the perspective of the externalities of highway infrastructures to its adjacent areas, the improvement of the accessibility of a certain area by the development of highway infrastructures leads to an increase in the price of housing. Hedonic pricing model or difference-in-differences estimators are popularly used methods to estimate the increase in housing prices near highway infrastructures [22–24]. With the accumulation of panel data and the improvement of computing power, such estimation become mature and popular and has been considered in policy decision making [11].

### 2.3. Evaluation of Ecological Externalities

From the ecological perspective, many studies focus on pollution emissions and noise of highway infrastructures [25], the damage of highway infrastructures to the ecology, and the countermeasures through the natural landscape, highway forest belts [22], and noise barriers [25]. Air pollution caused by the emission of pollutants by vehicles is a direct impact of highway infrastructures on the environment and is a serious problem [17]. Through empirical model and analysis, Yu and Zhou [26] investigated the effect of governmental highway spending on vehicle emissions. The results revealed that improved fuel efficiency and road conditions can cause more traffic. The elasticity of passenger emissions to highway spending is only one-fourth of that in the freight sector. Relying on a bottom-up transportation model, Mangones et al. [27] examined the effect of expanded highway capacity on traffic-related emissions of five pollutant criteria (CO, $NO_x$, PM10, $SO_2$, and VOC) in Bogotá, Colombia. The results suggested that adding capacity to the heavily congested road network of Bogotá could reduce traffic-related emissions immediately after the new roads start operating. Sovacool et al. [28] estimated the hidden social and environmental costs of transport externalities which are about $13.018 trillion per year globally.

Major emissions during transport are not only key sources of global climate change, but also harmful for the health of nearby residents [26]. From the perspective of the adjacent areas of highway infrastructures, Hamersma et al. [25] and Hamersma et al. [29] investigated residents' perception of live-ability change caused by negative environmental quality (i.e., noise, air pollution, and barrier effects) and the perception of accessibility change using questionnaire surveys and structural equation modeling.

Highways are not only likely to damage the natural environment, but also cause various ecological problems. Mansuroglu et al. [17] investigated a variety of negative externalities of highways on the natural resources including land consumption, removal of vegetation, and severance of agricultural areas by the road building process. Feng et al. [30] developed a before–during–after control-impact remote sensing (BDACI-RS) approach to quantifying the spatial and temporal changes of the environment during and after the construction of the Wujing Highway in China. The result showed that the impacts of the highway on the environment, which include vegetation and moisture conditions, degradation-recovery trends, land surface temperature, manifested the most in its proximity and faded away with distance. Cai and Lu [31] used remote sensing technology and GIS technology to quantitatively assess the road ecological effects. The results showed that the process of road construction increases the landscape fragmentation and decreases its contagion which affects animal migration. Ramísio et al. [10] estimated the highway runoff pollution based on a case study on 279-km Portuguese Highway. This study also used the willingness to pay (WTP) method to estimate the willingness of the public to afford the constructed wetlands cost.

### 2.4. Evaluation of Multiple Dimensional Externalities

Many studies on externalities of highway infrastructures do not strictly distinguish the dimensions of externalities. Using available data in Mexico and well-established methods, Cravioto et al. [13] calculated six categories of estimates of the externalities in Mexico. The results showed that road transport externalities amounted to at least US $59.42 billion per year or 6.24% of GDP in Mexico. By component, accidents represented the largest share (28%), followed by congestion (22%), greenhouse gases (21%), air pollution (13%), infrastructure (7%), and noise (9%). Moreover, Higgins et al. [32] investigated whether spatial trade-offs occur between the accessibility benefits of transportation and negative externalities from increased levels of harmful emissions and congestions in the single-detached property market around two highways in Hamilton, Canada. Using cross-sectional spatial-temporal hedonic models, the study revealed the evidence of a trade-off between transport advantage and environmental disadvantages in the study area. In addition, Nocker et al. [33] used the European accounting framework to assess and analyze external costs of a wide variety of transportation technologies within the life cycle of the road infrastructure. The European accounting framework allows to quantify and monetize impacts on public health, agriculture, and materials, but could not monetize ecological impacts. Moreover, this accounting framework allows calculating external costs of the transport infrastructures related to fuels, vehicles, and infrastructures.

The comprehensive literature review of the externalities of highway infrastructures in this study and a scientometric analysis of infrastructure externalities conducted by Zhu et al. [11] all showed that there are many research on the needs of transport, the impact of transport, and the externalities of transport, providing a solid scientific knowledge base for the evaluation of the externalities of highway infrastructures. However, there are still research gaps in two main perspectives. From a macro and holistic perspective, a human-dominated planet is a kind of social–economic–natural complex ecosystem dominated by human behaviors, sustained by a natural life support system, and vitalized by ecological processes [34]. The development of highway infrastructures comes from the needs of the economy, society, and environment, and conversely, has impacts on these three aspects. The lack of externality evaluation of any dimension will not only bias the view of highway-related externality problems, but also lead to the distortions in the planning and

development of highway infrastructures, leading to government failure or market failure for the governance of externalities [11,28,35]. The evaluation indexes in previous research are not comprehensive. On one hand, most of the previous studies only focused on the evaluation of one or two dimensions of the externalities of highway infrastructures. On the other, studies on the evaluation of the social and ecological externalities of highway infrastructures mainly focused on the negative externalities; studies on the evaluation of economic externalities of highway infrastructures mainly focused on the positive externalities. From a micro and technical perspective, due to the different forms of the externalities of highway infrastructures, there are few studies to quantify them with unified dimensional indicators, decreasing the comparability of evaluation results. Even quantifying and monetizing externalities of highway infrastructures, there are a lot of uncertainties and subjective treatments in the evaluation process. The ultimate purpose of the evaluation of the externalities of highway infrastructures is not only to reveal the impacts of all aspects of externalities, but also to reveal how to balance the impacts of all aspects of externalities, achieving maximum social welfare. To fill the above research gaps, this study aimed to make a comprehensive evaluation of the externalities of highway infrastructures from the economic, social, and ecological dimensions using the entropy weight method and grey correlation analysis, capturing the temporal and spatial trends of the externalities of highway infrastructures for a region.

## 3. Research Methodology

Comprehensive evaluations usually follow some principles including scientificity, objectivity, comparability, and feasibility [36]. Following these principles, the comprehensive three-dimensional evaluation of the externalities of highway infrastructures was achieved in three steps. The first step is to establish a three-dimensional evaluation index system through a comprehensive literature review and expert interviews. As a multi-index evaluation problem, the second step is to determine the relative weight of each index using the entropy weight method. The third step is to develop the comprehensive evaluation model by the grey correlation analysis.

### 3.1. Construction of Three-Dimensional Evaluation Index System

The construction of the three-dimensional evaluation index system of the externalities of highway infrastructures is from a perspective of a region. There are two reasons. First, the development of highway infrastructures serves more for a region [36]. The research on its externalities, such as the stimulus for economic development, air pollution, and the social welfare of a region's population, is also considered from the regional level [13]. In addition, most statistics are more oriented at the regional level. The selection of critical indicators follows three principles. First, the selection of the indicators should be systematic. The second principle is that the indicators are frequently used or widely acknowledged by the academy or the industry. The third principle is that there are available statistical data for each indicator in China.

Systematicness of the indicator selection is ensured from two aspects. The first aspect considers the aim of highway infrastructure development that is to support social development and improve the quality of life of human beings. The selection of the indicators first referred to some well-influential social indicators or indicators for quality-of-life and well-being. They are the Human Development Index (HDI), Social Progress Index (SPI), Gallup-Healthways Well-Being Index, and Canadian Index of Well-being (CIW) [37]. By referring to these social indexes and the characteristics of highway infrastructures, the basic medical care and personal safety that represent the basic human needs, the access to basic knowledge, health and wellness, and ecosystem sustainability that represent the foundations of wellbeing, and the freedom of travel and access to advanced education that represent the opportunity were included in the evaluation of the externalities of highway infrastructures. When selecting the indicators, this study also referred to the OECD's better life index that was developed in 2011 and highly related to the quality of life mea-

surement [37,38]. Health, personal security, income and wealth, environmental quality, education and skills, jobs and earnings, and social connections are areas of concern related to highway infrastructures. In general, highway infrastructures support the social welfare system that includes the economy, medical care, education, employment, and environment. Jansen and Denis [39] and Parry and Bento [35] considered social welfare effects of pollutant emissions from road vehicles and congestion costs, respectively, to support assessing various externality-related policy measures and taxes.

The second aspect to ensure systematicness is the synergy of social, economic, and ecological systems. This study adopted the theory of social–economic–natural complex ecosystem that emphasizes the synergy of social, economic, and ecological systems in a specific region [34]. The economic system includes production, consumption, reduction, and transportation. The social system includes technology, institution, and culture. The ecosystem concerns more with environmental sustainability and biodiversity [40]. To incorporate the main principles of sustainable development in transport planning, Sdoukopoulos et al. [36] also summarized the most commonly used indicators of the externalities of transport infrastructures from the perspectives of society, economy, and environment. It is well acknowledged that most measures of well-being correlate moderately with each other [41]. Considering the above, this study first divided the externalities of highway infrastructures into three dimensions: social, economic, and ecological externalities. Then, this study divided the externalities into positive and negative.

To ensure the realization of the second principle, this study extended the literature review. This study concurrently used the keywords of "externalit*", "transport*", and "evaluation" to perform an article search under all fields using WoS without time limitation. After reading the abstract of the searched articles, this study finally selected 38 articles that are most relevant to the externality evaluation of transportation infrastructures. This study first selected a set of preliminary evaluation indicators through reading all selected articles in this round and the previous round, counting the cumulative frequency of each indicator in the literature, and theoretically analyzing the rationale of the indicator as well as referring to the handbook on the external costs of transport that released by the European Commission [40]. Then, this study conducted interviews with experts in the transportation field to check the applicability of the indicators. In addition, this study checked the availability of relevant statistical data. Finally, the three-dimensional evaluation index system was constructed, as shown in Table 1.

(1) Evaluation indicators in the social dimension

The social externalities of highway infrastructures are often expressed as the quality of people's life and human health [34,42]. Based on the previous research and the social welfare system, this study divided the positive social externalities of the highway infrastructures into four aspects: transportation, education, medical care, and employment.

In terms of transportation, the top-mentioned externality is accessibility. The increased road network density and improved structure of the road network lead to the convenient, comfortable, and safe travel of residents [13]. The road network density was selected as an indicator of transportation level because it is highly related to mobility and accessibility. A well-connected, highly mobile, and reliable transportation system can shrink space within a given time [15]. Sdoukopoulos et al. [36] found that road network length per built-up area is a prevailing indicator to represent the social externalities of transport infrastructures. In terms of education and medical care, the development of the highway infrastructures not only facilitates the construction of educational institutions and medical care institutions, but also makes it more convenient for some regions to obtain better educational resources, medical resources, and employment opportunities [15,32]. Higgins et al. [32] used household access to regional employment via the road network as an indicator of accessibility. Because there are differences in size for educational institutions and medical care institutions, this study used the number of teachers serving ten thousand students and the number of beds in hospitals and health centers for ten thousand persons to represent education and medical care improvements, respectively. In terms of employment,

many studies revealed evidence of employment growth influenced by annual growth in the provision of major highways [7,43]. The convenience of transportation and the reduction of transportation costs make people obtain more employment opportunities [44]. Jiwattanakulpaisarn et al. [43] investigated the causal relationship between highway infrastructures and employment within the U.S. using the panel data over the period 1984–1997. The results revealed the evidence that employment growth is temporally influenced by annual growth in the provision of major highways within the same state and all other states. The proportion of employed persons in a region including urban and rural areas was selected as an indicator of employment improvement.

**Table 1.** Evaluation index system of externalities of highway infrastructures.

| Type | Dimension | Variable | Index | Code | Unit | Index Meaning | References |
|---|---|---|---|---|---|---|---|
| Positive externality | Social | Transportation | Road network density | PS1 | 1/km | Road length of municipal district divided by built-up area | [13,36,45] |
| | | Education | Education improvement | PS2 | person | Number of teachers serving ten thousand students | [1,36,46] |
| | | Medical care | Medical care improvement | PS3 | pcs | Number of beds in hospitals and health centers per ten thousand persons | [1,36,46] |
| | | Employment | Employment improvement | PS4 | % | Proportion of employed population | [7,43,44] |
| | Economic | External economic relations | External economic relations | PE1 | ten thousand dollars | Total import and export of goods and foreign direct investment per capita contributed by highway | [15,21,28] |
| | | Internal economic driving | Internal economic driving | PE2 | thousand dollars | PPP per capita | [15,36,47–50] |
| | | Affordability | Traffic costs | PE3 | yuan/(km·pcs) | Toll revenue divided by road length and number of civilian cars | [4,12,36,39] |
| | Ecological | Natural landscape | Afforestation coverage rate | PEO1 | % | Afforestation coverage area divided by built-up area | [22,42] |
| Negative externality | Social | Transportation | Traffic congestion | NS1 | pcs | Number of civilian cars divided by thousand $m^2$ road area | [13,39,51,52] |
| | | | Traffic accident losses | NS2 | yuan | Direct losses from traffic accidents per capita | [13,28,39,51,52] |
| | Ecological | Ecological environment | Air pollution | NEO1 | kg | Exhaust emission and smoke (dust) emission per capita | [13,17,29,32,36,39,44,53,54] |
| | | | Noise pollution | NEO2 | dB(A) | Noise detection level | [13,17,29,36,39,44,55] |

In terms of transportation, another top-mentioned externality is mobility. It is well accepted that mobility is often closely linked to one's independence, well-being, and quality of life [56]. In recent years, the transportation industry in Europe, America, Japan, and other countries has put forward the concept of mobility. For example, New York Government [57] proposed the goal of efficient mobility in OneNYC2050 plan. The goal of efficient mobility includes congestion-mitigation, safety, health, and sustainability. If the mobility brought by the transport is efficient, it can bring many benefits such as psychological benefits, exercise benefits, and community benefits [56]. Research has tested that the association between efficient transport mobility and quality of life is significant [56]. However, due to the rapidly increasing in car ownership and the concerns of global warming, negative externalities brought by highway infrastructures, such as congestion, accidents, air pollution, and noise, are widely criticized. In the realm of mobility, the increasing severity of traffic congestion and the increasing number of accidents have increased the transportation cost and safety

cost of residents [51,52]. For example, traffic congestion costs the United Kingdom economy an estimated £6.9 billion a year in lost time in 2019 [58]. Considering the safety and sustainability of road transport, pedestrians and cyclists are highly recommended [57,59]. The essential problem of congestion is the time lost [13,52]. However, it is very difficult to measure the direct time lost and such statistical data is rare. It is widely accepted that the overload of highway infrastructures is the direct cause of traffic congestion. For example, Tscharaktschiew [52] used traffic density as an indicator when modeling the congestion of highways to determine highway speed limits. Therefore, this study used the number of civilian cars per thousand square meter road area to reflect the level of overload of highway infrastructures. In addition, various social costs from road traffic accidents are frequently used indicators for the accidents of highway infrastructure [13]. Sdoukopoulos et al. [36] found that the number of road fatalities per built road area is a prevailing indicator to represent the safety-related externalities of transport infrastructures. This study used the traffic accident losses per capita as an indicator to monitor the changes of accidents of highway infrastructures.

(2)    Evaluation indicators in the economic dimension

The impacts of highway infrastructures on the economy take place via changes in generalized transport costs, accessibility, and mobility. Economic externalities refer to the external effects of the construction and operation of highway infrastructures on regional economic development, including external economic relations, internal economic driving, and labor productivity [60]. In terms of external economic relations, highway infrastructures constantly improve transportation efficiency, save transportation costs, shorten the space-time distance between production factors, promote urban extension and industrial diffusion, and provide fast access for commodity export [15]. Taking China for an example, the total freight volume of highway infrastructures accounts for about 60% of the total freight volume. To measure the improvement in the external economic relations, the total import and export of goods and foreign direct investment per capita were selected as the indicator of external economic relations. It is worth noting that various modes of transportation together contribute to the improvement of the external economic relationship. However, neither the government nor the institutions will release the contribution of each mode of transportation. This study adjusted this indicator by multiplying the proportion of cargo turnover of highway infrastructures in the total cargo turnover of various transportation modes to more accurately represent the contribution of highway infrastructures to external economic relations. Cargo turnover whose unit is a ton·kilometer considers not only weight, but also distance, representing the mobility ability of different transport modes [28].

In terms of internal economic driving, highway infrastructures actively promote industrial agglomeration and enterprise agglomeration within a region, promote the flow of labor force and the gathering of productivity, and improve the total amount of investment in a region [15]. Many empirical studies estimated the spillover effects of transportation investments on the gross domestic product (GDP) of national, state, or county levels [47] or the spillover effects on productivity [47,48,60]. Using a panel dataset of countries worldwide throughout 1996–2000 and the pooled mean group estimator, Calderón et al. [49] found that the contribution of infrastructure to the GDP ranged between 0.07 to 0.10. Jones [50] used the increase in per capital monthly household income as an indicator to measure the well-being-related externalities of energy efficiency investments. Using 18 OECD countries during 1870–2009, Farhadi [60] evaluated the direct impact of transport infrastructures on both labor productivity and total factor productivity. Different from GDP, PPP that is purchasing power party can eliminate the differences in price levels between countries in different years. Thopil and Pouris [61] used PPP to convert the externality costs of non-renewable electricity generation in South Africa to US currency. Considering the above and to measure the true improvement of living standards of people, this study selected PPP, which is the GDP per capita in purchasing power parity U.S. dollars frequently used as a social progress indicator [37], as an indicator of internal economic driving.

In addition, this study used the conversion rates provided by OECD data [62] to convert GDP to PPP. This study did not include labor productivity as an indicator because PPP and labor productivity are highly correlated which may cause a double-counting problem.

There are three essential components of mobility that are travel time, costs, and safety [56,57,63]. Sdoukopoulos et al. [36] found that fuel prices, transportation-related taxes, and share of household income devoted to transport are the most commonly used indicators of travel affordability. In terms of fuel prices, the impact of highway infrastructures on them is not decisive. However, the vehicle toll charged by the government is closely related to highway infrastructures [4,12]. This study proposed the unit traffic costs that are toll revenue divided by road length and number of civilian cars to represent the travel affordability of highway infrastructures. It must be pointed out that this indicator is influenced by statistics. Taking China as an example, the data of this indicator is not many because the statistics of toll highways and release of relevant data began in 2014. In addition, such statistics are only for provinces and the country. However, this indicator should not be overlooked and will contribute to the evaluation in the future because this statistical system has been formed already and is an ongoing statistic.

It is worth noting that local externalities caused by highway infrastructures, such as the effects of highway development on nearby housing prices [64], are not considered in this study. Although numerous articles discuss the impact of transportation infrastructures on the house prices caused by the commuting convenience or accessibility [11,65], from a region's perspective, such a kind of change in house price is more the relative change of house prices in different areas in the region. From the perspective of a region, the improvement of transportation infrastructure improves the image of the region and promotes its economic growth. The two most important factors which are economy and population affect the price of real estate.

(3)　Evaluation indicators in the ecological dimension

The ecological externalities are often expressed as environmental damage, disruption of ecosystem equilibrium, air pollution, and traffic noise [12,42]. First, with its expansion, highway infrastructure construction occupies more and more land, leading to environmental damage [17]. The best sites for highway development usually tend to be ideal for agriculture because they are flat and stable [17]. The removal of vegetation leads to changes in ecology and soil and runoff pollution [10]. Sdoukopoulos et al. [36] found that area taken by transport infrastructures is one of the most commonly used indicators of the evaluation of the externalities of transport infrastructures. It is a pity that this indicator generally lacks statistics of relevant data. To governance ecological externalities, the government develops some natural landscapes [42], highway forest belts [22], noise barriers [25], and wetlands [10] along highway infrastructures. On the contrary, these related infrastructure produces positive external effects on the environment. Second, highway infrastructures also cause many air pollution emissions, such as particulate matter 2.5 microns (PM 2.5) and $CO_2$ [13,32,53]. Emission data is a frequently used indicator for air pollution of highway infrastructures [13,32,54]. There are available statistics data for exhaust emission and smoke emission in China. Third, highway infrastructures are characterized by large flow and fast speed, which also produce a lot of noise pollution [25,29]. Noise is one of the most obvious impacts of highway infrastructures. However, its effects are often given lower priority than economic or other environmental impacts because they are rarely visible and difficult to be monetarily quantified [17]. There are only a few research using the marginal cost principle to quantify the noise charges of transport infrastructures [66]. The level of average constant noise from road traffic is an often-used indicator [13].

*3.2. Determination of Index Weight Using an Entropy Weight Method*

This study adopted the entropy weight method to determine the relative weight of each index. Compared with various subjective weighting methods, the entropy weight method can avoid the interference of human factors on the weight of an indicator, enhancing

the objectivity of the comprehensive evaluation results. The entropy weight method has been widely used in determining the relative weight of criteria [67–70].

In information theory, the entropy of a random variable is the average level of "information", "surprise", or "uncertainty" inherent to the variable's possible outcomes [71]. The informational value of a message from a variable depends on the degree to which the content of the message is surprising. If a highly likely event occurs, the message carries very litter new information. On the contrary, the message is much more informative if a highly unlikely event occurs. Based on this logic, the entropy weight method evaluates the amount of information provided by an index through the dispersion of the index data. The higher the degree of dispersion of the measured value, the higher degree of differentiation of the index, and more information can be derived. Therefore, a higher weight should be given to the index [68].

In this study, the values of the evaluation indexes are all quantitative data that can be retrieved through annual statistical reports. However, the measurement units of various evaluation indexes are also different. The weight cannot be determined directly by the value. Moreover, along with many years' development of the highway infrastructure, the data of the multi-dimensional indexes has a large degree of dispersion. The degree of dispersion can well reflect the degree of the influence of the externalities of the highway infrastructure. Furthermore, if the data of an externality is relatively concentrated, the influence of the externality has reached a very common and more recognized level. It should not be given more weight because it is more likely to be internalized by the government or market. For example, the carbon trading market has emerged because $CO_2$ emissions have received more attention. Considering the above, the entropy weight method is suitable for determining the relative weight of the identified indexes in this study.

According to the entropy weight method [68,70,72,73] and the objective of this study, this study determined the weights of evaluation indexes through four main steps.

(1)  Construction of the index matrix

An evaluation problem with *m* valuation indexes and *n* statistical samples can be formulated in a matrix format $(R_{ij})_{m*n}$ as follows:

$$R_{ij} = \begin{pmatrix} r_{11} & \cdots & r_{m1} \\ \vdots & \ddots & \vdots \\ r_{1n} & \cdots & r_{mn} \end{pmatrix} \tag{1}$$

$r_{ij}(i = 1, 2, 3 \dots, m; j = 1, 2, 3 \dots n)$ is the value of the index *i* in the sample *j*.

(2)  Standardization of the index matrix

To eliminate the impact of different dimensions on the evaluation results, it is necessary to standardize each index into a non-dimensional index. They are two types of indexes. For positive externalities, the larger value of the index the better; for negative externalities, the lower value of the index the better. The positive and negative indexes were standardized by Equations (2) and (3), respectively.

$$x_{ij} = \frac{r_{ij} - \min(r_i)}{\max(r_i) - \min(r_i)} \tag{2}$$

$$x_{ij} = \frac{\max(r_i) - r_{ij}}{\max(r_i) - \min(r_i)} \tag{3}$$

To avoid the distortion by zero values in the standardized index matrix [68], this study made a non-negative transformation using the following Equation (4).

$$x'_{ij} = x_{ij} + \min(x_{ij}) + 0.01 \tag{4}$$

(3)   Calculation of the entropy values of the indexes

This study calculated the probability of the occurrence of each sample for each index using the proportion of each sample in the total value of the corresponding index, as shown in Equation (5). Then the entropy value of each index was calculated using Equation (6).

$$a_{ij} = \frac{x'_{ij}}{\sum\limits_{j=1}^{n} x'_{ij}} \tag{5}$$

$$b_i = -\frac{1}{\ln n} \sum_{j=1}^{n} a_{ij} \ln a_{ij} \tag{6}$$

(4)   Determination of the relative weight of each index

The relative weight of each index is calculated using the following equation.

$$w_i = \frac{1 - b_i}{m - \sum\limits_{i=1}^{m} b_i} \tag{7}$$

As this study differentiated the positive and negative externalities of highway infrastructure, the set of the relative weights of the indexes for positive externalities were separated from that of the negative externalities. The final weight vector of the indexes is as follows.

$$W = (w_1, w_2, w_3, \ldots, w_m)^\tau \tag{8}$$

*3.3. Comprehensive Evaluation Using Grey Correlation Analysis*

This study developed the comprehensive evaluation model using the grey correlation analysis. Grey correlation analysis proposed by Deng [74] is an approach to quantitative analysis of dynamic process using similarity of trend and pattern between the reference sequence and comparative sequences. Grey correlation analysis evaluates the relationship between the reference sequence and comparative sequences according to the similar degree of the geometric shape of the sequences curve [75]. The closer the curves, the greater the correlation among the sequences and the larger the correlation degree. In addition, the grey approach can work well with irregular data. The grey correlation analysis has been applied in various fields of engineering and management [73,75,76]. Li et al. [73] used this method to investigate the nonlinear multiple-dimensional model of the socio-economic activities' impact on the air pollution of Beijing.

To evaluate the impact of the multi-dimensional externalities of highway infrastructure on the social, economic, and ecological systems in the dynamic development process of the highway infrastructure, the grey correlation analysis is a suitable method to be chosen. Moreover, the grey correlation degree can transform the multi-dimensional values of the externality indexes into one-dimensional measurement value, realizing the comprehensive evaluation and exploring the development trend of the externalities of highway infrastructure.

Integrating with the relative weights of indexes determined by the entropy weight method, the comprehensive evaluation model of the externalities of highway infrastructure was established through the following five steps.

(1)   Determination of the reference index set

Based on the evaluation index matrix $(R_{ij})_{m*n}$, let $R^* = [r_1^*, r_2^*, \ldots, r_m^*]$ represents the reference sequence where $r_i^* (i = 1, 2, \ldots m)$ is the reference value of the index *i*. $R^*$ was constructed by selecting the optimal value of each evaluation index. For positive externalities, the optimal value is the largest value; for negative externalities, the optimal value is the lowest value. Therefore, the evaluation results of this study using the grey

correlation analysis reveal the closeness to the optimal condition. Let $R_j = [r_{1j}, r_{2j}, \ldots, r_{mj}]$ represent the comparative sequence of the sample $j$.

(2)    Standardization of the index matrix

In the application of grey correlation analysis, dimensionless data processing needs to be carried out to transform all data into a unified measurement scale. Using Equations (2) and (3), and combining the reference and comparative sequences, the standardized index matrix $X'$ was obtained.

$$X^1 = \begin{pmatrix} x_1^* & \cdots & x_m^* \\ x_{11}' & \cdots & x_{m1}' \\ \vdots & \ddots & \vdots \\ x_{1n}' & \cdots & x_{mn}' \end{pmatrix} \tag{9}$$

(3)    Determination of the differences

The differences between the comparative sequence of the sample $j$ and the reference sequence can be calculated as follows.

$$\Delta x_{ij} = \left| \left| x_{ij}' - x_i^* \right| \right| \tag{10}$$

(4)    Determination of the extreme values

The maximum and minimum values of the different sequences are calculated using the following equations.

$$\Delta x_{\max} = Maximum \Delta x_{ij} \tag{11}$$

$$\Delta x_{\min} = Minimum \Delta x_{ij} \tag{12}$$

(5)    Calculation of the grey correlation coefficient

Grey correlation coefficient reflects the development trend of the difference between the reference sequence and comparison sequence curves. The difference is the degree of correlation. The grey correlation coefficient of each index in each sample is calculated using the following equation

$$\varepsilon_{ij} = \frac{\Delta x_{\min} + \rho \Delta x_{\max}}{\Delta x_{ij} + \rho \Delta x_{\max}} \tag{13}$$

where $\rho$ is the discrimination coefficient, $\rho \in (0, 1)$. The smaller $\rho$ value means the stronger the discrimination ability. Generally $\rho = 0.5$ [73,75].

(6)    Calculation of the comprehensive evaluation value

Based on the relative weights and the grey correlation coefficients, the comprehensive evaluation value for the sample $j$ can be obtained using the following equation:

$$T_j^+ = \sum_{i=1}^{m_1} w_i \varepsilon_{ij} \tag{14}$$

$$T_j^- = \sum_{i=1}^{m_2} w_i \varepsilon_{ij} \tag{15}$$

$$T_j = T_j^+ + T_j^- \tag{16}$$

where $m_1$ and $m_2$ represent the numbers of indexes for positive externalities and negative externalities, and $m_1 + m_2 = m$; $T_j^+$, $T_j^-$, $T$ denote the comprehensive evaluation values of the positive externalities, negative externalities, and total externalities.

## 4. Empirical Analysis Results

This study conducted an empirical analysis using the statistical data from the Jiangsu province of China to verify the feasibility of the comprehensive evaluation model. Jiangsu

province has become one of the provinces with the highest comprehensive development level in China. Jiangsu covers an area of 107,200 square kilometers and has a population of 84.8 billion located in 13 cities. According to the statistical data from the National Bureau of Statistics [77] of China, the GDP of Jiangsu province reached 1619 billion US dollars in 2020 and ranked second in China. The highway mileage of Jiangsu province accounts for about 5% of the total highway mileage in China.

*4.1. Data Source of Indexes*

The basic data of the externality indexes of highway infrastructures in Jiangsu province of China was obtained from China Statistical Yearbook, Statistical Bulletin of National Economic and Social Development in Jiangsu Province, and Statistical Yearbook of Jiangsu Province from 2008 to 2020. This study collected two sets of statistical samples. From the temporal perspective, one set of samples collected the data of externality indexes of highway infrastructures in Jiangsu province from 2008 to 2020. From the spatial perspective, another set of samples collected the data of externality indexes of highway infrastructures in different cities of Jiangsu province in 2020. The data sets are shown in Tables 2 and 3. Considering the insufficient statistical data of the indicator of affordability (PE3) (only 2015–2020), this study did not include it in this empirical analysis. However, this indicator can adjust the evaluation in the future with the accumulation of statistical data in China. Regions having enough data can include this indicator in their evaluations. This study conducted all analyses using Microsoft excel.

**Table 2.** Data of the externality indexes of highway infrastructures in Jiangsu from 2008 to 2020.

| Year | PS1 | PS2 | PS3 | PS4 | PE1 | PE2 | PEO1 | NS1 | NS2 | NEO1 | NEO2 |
|---|---|---|---|---|---|---|---|---|---|---|---|
| 2008 | 9.90 | 609.00 | 28.80 | 60.56% | 2642.44 | 12.56 | 42.63% | 7.88 | 0.65 | 5.00 | 69.0 |
| 2009 | 9.85 | 637.00 | 30.40 | 60.52% | 3121.94 | 14.05 | 42.00% | 9.15 | 0.70 | 4.91 | 68.5 |
| 2010 | 9.75 | 651.00 | 31.45 | 60.04% | 3875.25 | 15.86 | 42.07% | 10.57 | 0.63 | 4.84 | 68.5 |
| 2011 | 9.30 | 668.00 | 34.60 | 59.20% | 4099.75 | 17.44 | 42.12% | 11.79 | 0.75 | 4.52 | 67.9 |
| 2012 | 9.57 | 678.00 | 38.80 | 58.75% | 4214.79 | 18.68 | 42.17% | 13.02 | 0.89 | 4.53 | 68.4 |
| 2013 | 9.71 | 692.00 | 42.98 | 58.49% | 4315.74 | 19.86 | 42.44% | 14.25 | 0.83 | 4.56 | 68.0 |
| 2014 | 9.72 | 680.00 | 45.84 | 58.12% | 4778.31 | 20.94 | 42.60% | 15.52 | 0.76 | 4.37 | 67.2 |
| 2015 | 9.73 | 672.00 | 48.26 | 58.12% | 7723.45 | 22.18 | 42.83% | 16.63 | 0.80 | 3.94 | 67.9 |
| 2016 | 10.47 | 669.00 | 51.89 | 57.87% | 7396.85 | 23.23 | 42.94% | 17.99 | 0.76 | 3.84 | 67.9 |
| 2017 | 10.64 | 668.00 | 54.60 | 57.85% | 7812.27 | 24.43 | 42.96% | 19.66 | 0.69 | 5.12 | 68.0 |
| 2018 | 10.53 | 667.00 | 59.74 | 57.86% | 9246.48 | 26.13 | 43.14% | 20.93 | 0.76 | 4.72 | 67.5 |
| 2019 | 10.55 | 670.00 | 59.71 | 57.90% | 10,700.54 | 27.72 | 43.39% | 21.89 | 0.71 | 5.03 | 67.5 |
| 2020 | 10.62 | 674.00 | 58.90 | 57.72% | 12,561.18 | 28.96 | 43.47% | 22.57 | 0.67 | 4.98 | 66.8 |

**Table 3.** Data of the externality indexes of highway infrastructures in cities of Jiangsu in 2020.

| City | PS1 | PS2 | PS3 | PS4 | PE1 | PE2 | PEO1 | NS1 | NS2 | NEO1 | NEO2 |
|---|---|---|---|---|---|---|---|---|---|---|---|
| Nanjing | 10.75 | 691 | 62.40 | 51.98% | 11,518.31 | 38.06 | 44.69% | 16.41 | 0.38 | 3.38 | 67.7 |
| Wuxi | 11.40 | 680 | 61.70 | 56.11% | 9309.65 | 39.62 | 43.43% | 29.62 | 0.38 | 3.44 | 67.3 |
| Xuzhou | 9.25 | 616 | 60.20 | 53.07% | 4156.46 | 19.27 | 43.10% | 34.43 | 0.63 | 4.61 | 67.4 |
| Changzhou | 10.18 | 657 | 51.70 | 56.92% | 5467.92 | 35.34 | 43.27% | 27.07 | 0.59 | 5.02 | 66.1 |
| Suzhou | 15.42 | 650 | 56.40 | 58.65% | 18,493.14 | 37.86 | 43.10% | 38.61 | 0.33 | 2.85 | 66.7 |
| Nantong | 11.74 | 725 | 61.30 | 62.91% | 5564.10 | 31.03 | 43.29% | 31.86 | 0.80 | 4.35 | 65.3 |
| Lianyungang | 8.91 | 704 | 57.50 | 54.99% | 2708.88 | 17.03 | 42.29% | 28.91 | 0.94 | 5.39 | 64.0 |
| Huaian | 10.42 | 726 | 61.30 | 58.98% | 3576.81 | 20.90 | 42.60% | 16.76 | 1.07 | 5.37 | 63.9 |
| Yanchegn | 9.46 | 723 | 61.70 | 62.29% | 1207.80 | 21.20 | 43.60% | 32.19 | 1.17 | 4.16 | 66.0 |
| Yangzhou | 9.49 | 732 | 51.00 | 59.70% | 2389.52 | 31.72 | 44.67% | 29.31 | 0.75 | 4.93 | 67.0 |
| Zhenjiang | 10.68 | 724 | 44.90 | 62.10% | 3907.47 | 31.43 | 43.38% | 25.95 | 0.77 | 6.57 | 65.6 |
| Taizhou | 9.79 | 757 | 64.10 | 61.68% | 5181.09 | 28.08 | 42.57% | 28.29 | 0.79 | 4.65 | 64.2 |
| Suqian | 9.38 | 544 | 62.10 | 56.77% | 2583.50 | 15.65 | 44.99% | 37.45 | 0.89 | 3.97 | 64.9 |

### 4.2. Relative Weights of Indexes

Based on the statistical data of Jiangsu province from 2008 to 2020 and the proposed entropy weight method, this study calculated the relative weight of each externality index for evaluating the positive and negative externalities of highway infrastructures in Jiangsu province of China, as shown in Table 4.

**Table 4.** Weights of the externality indexes.

| Type | Dimension | Weight of Dimension | Code | Weight of an Index in Dimension | Comprehensive Weight of an Index |
|------|-----------|---------------------|------|----------------------------------|----------------------------------|
| Positive externality | Social | 0.5677 | PS1 | 0.1918 | 0.1089 |
| | | | PS2 | 0.0939 | 0.0533 |
| | | | PS3 | 0.2545 | 0.1445 |
| | | | PS4 | 0.4597 | 0.2610 |
| | Economic | 0.2803 | PE1 | 0.6158 | 0.1726 |
| | | | PE2 | 0.3842 | 0.1077 |
| | Ecological | 0.1519 | PEO1 | 1.0000 | 0.1519 |
| Negative externality | Social | 0.4505 | NS1 | 0.6526 | 0.2940 |
| | | | NS2 | 0.3474 | 0.1565 |
| | Ecological | 0.5495 | NEO1 | 0.6673 | 0.3667 |
| | | | NEO2 | 0.3327 | 0.1828 |

In the application of the entropy weight method, the greater entropy weight indicates a greater variation of the relevant index, revealing much more information and the influence of the externalities of highway infrastructures. As can be seen from the result, for the positive externalities, the four indexes with the highest comprehensive weights are the employment improvement (PS4), afforestation coverage rate of built-up area (PEO1), medical care improvement (PS3), and external economic relations (PE1); for the negative externalities, air pollution (NEO1) and traffic congestion (NEO2) are indexes with the highest comprehensive weights.

### 4.3. Grey Correlation Coefficient Results

Based on the statistical data and the proposed grey correlation analysis, this study calculated the grey correlation coefficient of each index for the externalities of highway infrastructures in Jiangsu province for the past 13 years and that in cities of Jiangsu province in 2020. From the temporal perspective, the grey correlation coefficient results were shown in Figure 1.

According to the principle of grey correlation analysis, when $\varepsilon_{ij} = 1$, an externality reaches the optimal value; when $\varepsilon_{ij} = 0$, there is no correlation between the optimal state of an externality, indicating that the externality is the worst in that year. As can be seen from Figure 1a, in the social dimension, the medical care improvement and road network density continuously keep getting better; on the contrary, the traffic congestion and employment improvement continuously keep getting worse. As can be seen from Figure 1b, in the economic and ecological dimension, except for air pollution, everything else keeps getting better.

### 4.4. Comprehensive Evaluation Results

Based on the relative weights and grey correlation coefficients, comprehensive evaluation results were obtained. The trends of related externalities of highway infrastructures in Jiangsu province in the past 13 years were shown in Figure 2. The status of the related externalities of highway infrastructures in 13 cities of Jiangsu province in 2020 was shown in Figure 3.

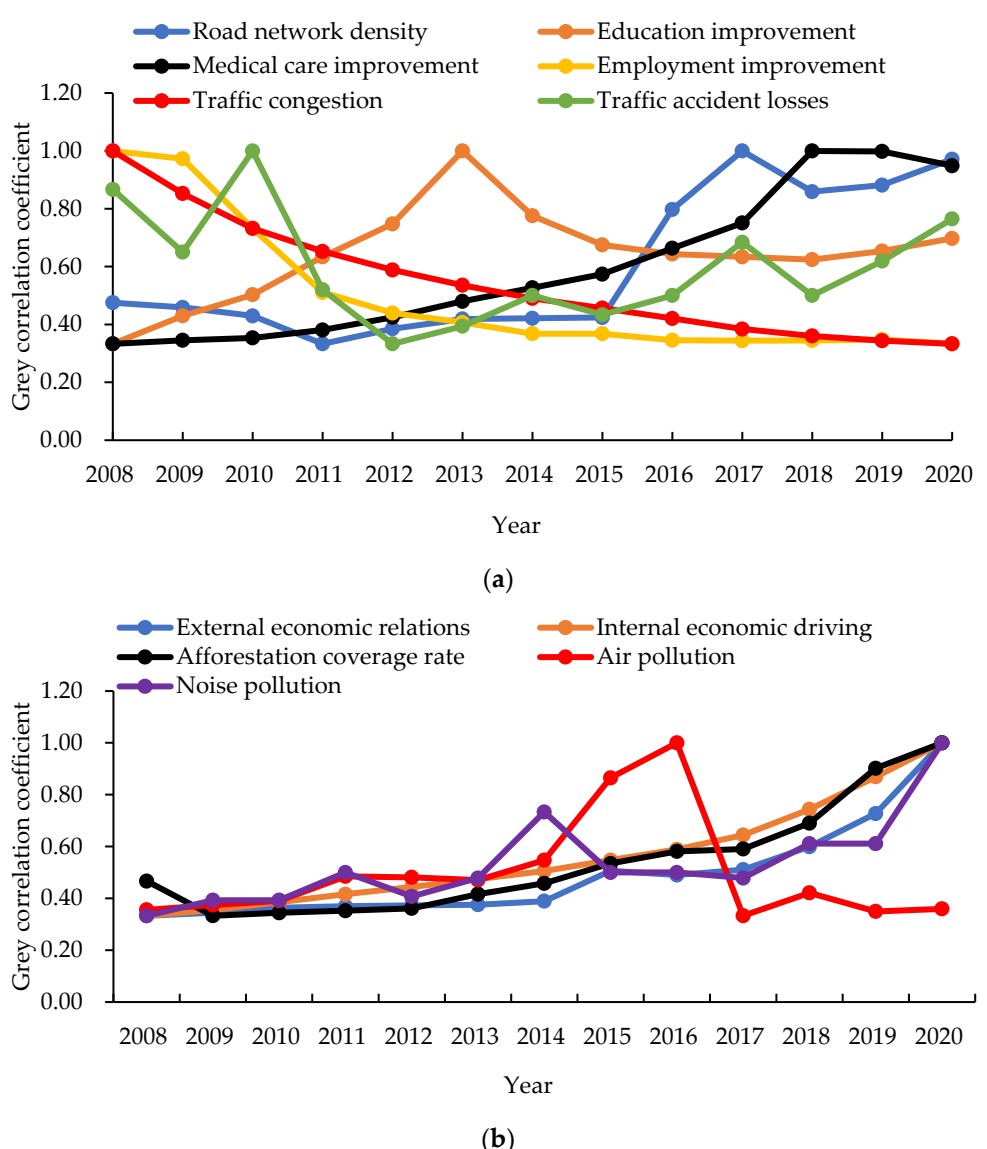

**Figure 1.** Grey correlation coefficient results in Jiangsu province for the past 13 years. (**a**) Social dimension; (**b**) economic and ecological dimensions.

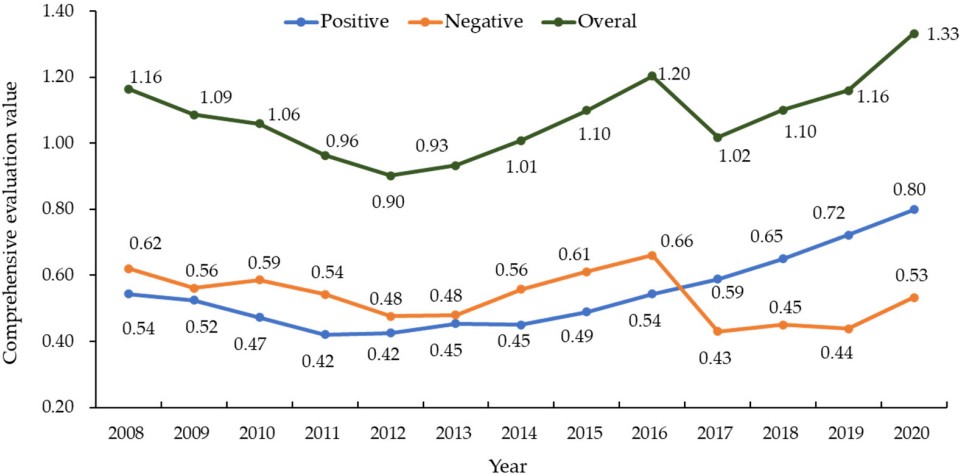

**Figure 2.** Trends of externalities of highway infrastructures in Jiangsu province in the past 13 years.

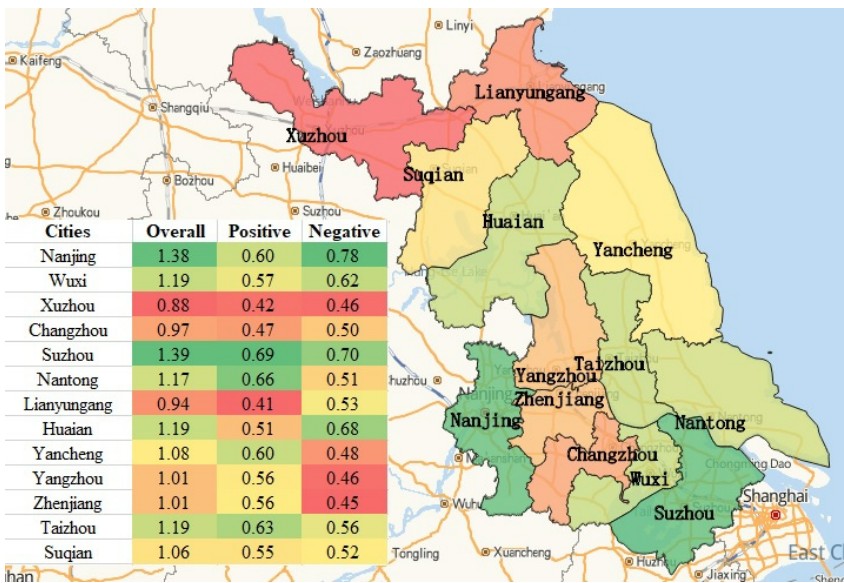

**Figure 3.** Status of externalities of highway infrastructures in the 13 cities of Jiangsu province.

As can be seen from Figure 2, the overall development trend shows a trend of decreasing first and then increasing gradually. There is a huge turning point in 2016–2017 which is mainly caused by the change of the negative externalities.

As can be seen from Figure 3, through the comprehensive evaluation of the externalities of highway infrastructures in 13 cities in 2020, Nanjing and Suzhou were identified as the cities with the highest evaluation value; Xuzhou and Lianyungang were identified as the cities with the lowest evaluation value. Taking Suzhou and Xuzhou which have extreme values as an example, both positive externalities and the well-control of negative externalities contribute to the overall value of Suzhou; while the total score of Xuzhou is very low due to the improper governance of both positive and negative externalities.

## 5. Discussions and Implications

Through the empirical analysis, it can be concluded that the proposed comprehensive evaluation model is feasible and effective in evaluating the externalities of highway infrastructures. The model plays a prominent role in the following three aspects.

### 5.1. People's Needs and Social Development

The proposed model successfully captures the value of externality-related information. Because highway infrastructures are important support for social development, the relative weights reflect people's needs and the development of a society in the past 13 years. As previously explained, the relative weights from the entropy weight analysis indicate the amount of information provided by an index through the dispersion of the index data. The results in Table 4 indicate that great changes have taken place in the following aspects under the joint action of highway infrastructures and other infrastructures.

From the perspective of positive externalities, great changes have taken place in the employment improvement (PS4), afforestation coverage rate of built-up area (PEO1), medical care improvement (PS3), and external economic relations (PE1). The significant positive effect of highway infrastructures on the economy has always been recognized [14,78]. In addition, the relevant literature on the development of Jiangsu province shows that there was an increasing demand for medical care and ecological environment protection. The government has issued many medical care-related policies and taken many related measures [79,80]. Compared with the continuously high level of education, the medical care conditions in Jiangsu province have been greatly improved. What is more, in 2005, the General Secretary of China Xi Jinping pointed out that "Lucid waters and lush mountains are invaluable assets". Jiangsu province has made great efforts in urban greening.

From the perspective of negative externalities, great changes have taken place in air pollution (NEO1) and traffic congestion (NEO2). According to the 2018 annual report of China Motor Vehicle Environment Management, China has become the largest country in motor vehicle production and sales in the world for nine consecutive years [81]. Motor vehicle pollution has become an important source of air pollution in China. The development of highway infrastructures has further promoted the increase in motor vehicle ownership and more traffic [26]. The situation of traffic congestion in Jiangsu province is getting worse. "Slow driving, parking difficulty, crowded driving" has become the main problem of social concern.

### 5.2. Distance to Optimal Development State

The proposed model successfully captures the externalities of highway infrastructures from the perspective of the distance to the optimal state. According to the principle of grey correlation analysis, the larger the grey correlation coefficient is, the closer it is to the optimal state. The results in Figure 1 show the trend of getting better or worse in all aspects of the externalities of highway infrastructures.

From the social dimension, except for the afore-mentioned medical care improvement, the road network density has been getting better. Jiangsu province attaches great importance to road construction. A lot of policies, plans, and investments have been made [82]. In 2019, the total mileage of the Jiangsu highway has reached 4711 km. The road network density ranked first in China. In addition, many highway service stations were renewed and have become internet celebrity visiting places [83]. As two negative externalities, the traffic accident losses per capita started to continuously get better from 2013, while the traffic congestion has always been getting worse. The most stringent traffic regulations in history have been implemented since 2013 [84]. The reduction of traffic losses per capita is related to China's attention to safety management and stringent traffic regulations. As for employment improvement, this study obtained a similar conclusion with Li and Whitaker [15] that highway investment has a positive but limited effect on jobs.

From the economic and ecological dimensions, there is a huge turning point in air pollution in 2016–2017. Through in-depth analysis of the data, it is found that the road length in Jiangsu province achieved huge growth in 2016, with an annual growth rate of 10.43%, which is twice higher than the average annual growth rate of 5%. As a result, the annual growth rate of pollution emission in Jiangsu province in 2017 reached 33.81%, which is 33 times higher than the average annual growth rate of 1%. There is a long way for the whole society to get back to the trend of getting better. It is worth noting that, although the affordability of highway infrastructures was not included in this empirical analysis, it generally shows a positive trend because the unit travel cost was decreasing according to the statistical data, representing positive externalities of highway infrastructures. Along with the accumulation of related statistical data, the status of the economic dimension will be adjusted in the future.

### 5.3. Temporal and Spatial Trends

The proposed model can not only evaluate the trend of the externalities of highway infrastructures, but also evaluate and compare the status of the externalities of highway infrastructures in different regions. From the temporal perspective, the comprehensive evaluation results in Figure 2 show a trend of decreasing first and then increasing gradually. There is a lag of positive externalities of highway infrastructures. It may take many years for the highway infrastructure to produce significant positive external effects on society and the economy. However, the direct impacts caused by highway infrastructures, such as the negative impact to the environment because of the land consumption and removal of vegetation and the air pollution caused by cars, are immediate results of the construction of highway infrastructures and soon show up [17]. With the development of society and the appropriate measures taken by the government to deal with negative externalities of highway infrastructures, the overall trend has gradually improved. The sharp decline in

2017 illustrates two important points. First, the convenient, comfortable, and safe travel led by better conditions and longer length of highway infrastructures makes people travel more and farther, causing more air pollution. This conclusion is the same as Yu and Zhou [26] and Hamersma et al. [44] that the improved road condition can cause more traffic and more vehicle emissions. Second, the government failed to anticipate this result and make corresponding plans. In addition, the relevant authorities relaxed their vigilance against vehicle emissions. As this serious problem is common throughout the country, the Chinese government implemented control from two aspects which are new vehicle purchase tax and new regulations on "national six" exhaust emission standards from 2017. From the spatial perspective, the status of the governance of the externalities of highway infrastructures in 13 cities of Jiangsu province is clearly shown in Figure 3. It clearly showed that the status of the externalities of highway infrastructures varies depending on the geographic conditions. The government can take different governance measures and efforts according to the current situations in different regions.

The above conclusions have two important implications. First, the designer or the government authorities need to think about how to balance the relationships among the society, economy, and ecology when planning highway infrastructures. The planner or design should put the concept of the harmonious relationship between humans and nature, subsistence and development, and alteration and respect of nature into highway design [3,26]. Road builders' negligence of natural and cultural conditions during the planning stage of highways leads to a slew of environmental impacts [17].

## 6. Conclusions and Recommendations

To make a comprehensive evaluation of the externalities of highway infrastructures, this study developed a three-dimensional evaluation model of the externalities of highway infrastructures by a comprehensive of multiple methods including the literature review, expert interview, entropy weight methods, and grey correlation analysis. Statistical data of 13 years and 13 cities in Jiangsu province of China was used to validate the proposed model. The analysis results showed that the developed model is feasible and effective in evaluating the externalities of highway infrastructures as the analysis results are consistent with reality. In addition, the model can capture the value of externality-related information, the distance to the optimal state of the externalities of highway infrastructures, and the temporal and spatial trends of the externalities of highway infrastructures for a region.

The results of this study first pave the way for the research on the influencing factors and the formation mechanism of the multi-dimensional externalities of highway infrastructures. Moreover, the results provide theoretical support for the scientific formulation of relevant policies and decision-making for the government. Various stakeholders, such as policymakers, planners, and designers, can use this comprehensive evaluation model and its results to make more forward-looking, sustainable, and targeted decisions on the planning and construction of highway infrastructures, avoiding distortion problems. Although the objective of this study has been achieved, there are some limitations of this study. The foremost limitation stems from the fact that transportation is only one of the many factors that may account for the development of society and economy and the impact on the ecology. It is unrealistic to completely isolate transportation effects from the effects of other infrastructure systems. In addition, the selected indicators in this study are currently the most prevailing. In the future, new and significant indicators should be selected into the evaluation index system with the social development and the development of highway infrastructures. The comprehensive evaluation of the externalities of highway infrastructures can be thought of as the evaluation of the phenomenon or results. Based on the results, future research can focus on the influencing factors and their influencing mechanism of the externalities of highway infrastructures, which is good for effective and long-term governance of the externalities of highway infrastructures.

**Author Contributions:** Conceptualization, L.Z. (Lei Zhu), L.Z. (Lina Zhang), Q.Y. and J.D.; methodology, L.Z. (Lei Zhu), L.Z. (Lina Zhang), Q.Y. and X.Z.; data collection and formal analysis, L.Z. (Lei Zhu), L.Z. (Lina Zhang) and Q.Y.; qualitative discussions, L.Z. (Lei Zhu), L.Z. (Lina Zhang), and X.Z.; writing—original draft preparation, L.Z. (Lei Zhu) and L.Z. (Lina Zhang); writing—review and editing, L.Z. (Lei Zhu), L.Z. (Lina Zhang) and X.Z.; finalizing the MS word, L.Z. (Lei Zhu); supervision, J.D. All authors have read and agreed to the published version of the manuscript.

**Funding:** This work was supported by the Social Science Foundation of Jiangsu Province (21SHB010) and the National Natural Science Foundation of China (NSFC-71801038).

**Data Availability Statement:** Data generated or analyzed during the study are available from the corresponding author by request.

**Conflicts of Interest:** The authors declare no conflict of interest.

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
