# Peer review of "A Three-Dimensional Evaluation Model of the Externalities of Highway Infrastructures to Capture the Temporal and Spatial Distance to Optimal—A Case Study of China"

_buildings, doi:10.3390/buildings12030328_

Round 1

Reviewer 1 Report

  • The model proposed aims at evaluating the externalities of highway infrastructure. The model development is largely dependent on the available data which makes its application and adoption very limited. However, the main concern is that the authors fail to explain the relationship between the various externalities indicators chosen and the transportation infrastructure even though these externalities are widely accepted as externalities of transportation infrastructure by the academic community but these indicators can be externalities to other types of investments or policies.
  • The indicators chosen are highly correlated which may cause the problem of double-counting. The choice of indicators must be well-studied and justified which is amply not presented. For example, mobility and accessibility are considered in one indicator even though they are different.     

Author Response

Response to Reviewers

Dear Editors and Reviewers,

We would like to thank you for the positive and constructive comments and suggestions. We have substantially revised our manuscript according to the valuable comments. The point-to-point responses to the comments are listed as follows. The revisions made to the manuscript can be easily seen because the manuscript is using the “Track Changes” function. For references that were deleted and cannot be shown in the manuscript, the authors marked the corresponding places with “Yellow” color.

Response to Reviewer #1’s comments:

  • The reviewer commented that the model proposed aims at evaluating the externalities of highway infrastructures. The model development is largely dependent on the available data which makes its application and adoption very limited.

Response: First of all, thank you for asking this question. Comprehensive evaluations usually follow some principles including scientificity, objectivity, comparability, feasibility, etc. To ensure the feasibility, the available data is the precondition. For example, in the era of big data, the existence of data is the key. In addition, using available data, especially staticstical data, is also one of the widely accepted methods in the research community. In the future, if the authors could find other important variables that do not have statistical data, we will propose them to the government to improve the statistical system. But currently, we conducted our research based on available data. Although the authors followed the above principles to develop the comprehensive evaluation model, we forgot to emphasize them in the original manuscript. Thanks for the reviewer’s question, the authors added these principles at the beginning of the section “3. Research Methdology” in the revised manuscript.

  • The reviewer commented that the authors failed to explain the relationship between the various externalities indicators chosen and the transportation infrastructure even though these externalities are widely accepted as externalities of transportation infrastructure by the academic community but these indicators can be externalities to other types of investments or policies.

Response: The authors much appreciated the reviewer for raising this question, providing us an opportunity to improve our manuscript. The authors wanted to answer this question from three aspects.

First, the authors re-reviewed the pinciples and practice of establishing the index system. Previously, the authors refereed to the theories of social welfare system to ensure the systematicness of the indicator selection which makes the reviewer feel less systematic and specific. It is widely acknowledged that the ultimate aim of highway infrastructure development that is to support social development and improve the quality of life of human beings. The authors conducted another round of literature review about the social indicators. The authors found that the Human Development Index (HDI), Social Progress Index (SPI), Gallup-Healthways Well-Being Index, Canadian Index of Well-being (CIW), and the OECD’s better life index are well-influential social indicators. Comparing our developed index system and the above social indicators, we ensured that our index system is systematic. Our developed index system includes (1) the basic medical care, income and wealth, jobs and earnings, personal safety that represent the basic human needs, (2) the access to basic knowledge, health and wellness, and ecosystem sustainability that represent the foundations of wellbeing, and (3) the freedom of travel and the access to advanced education that represent the opportunity. The authors also adopted the theory of Social-Economic-Natural Complex Ecosystem that emphasizes the synergy of social, economic, and ecological systems when developing the index system. Accordign to these considerations, the authors revised the manuscript in the section of “3.1. Construction of Three-dimensional Evaluation Index System” to explain the systematicness of the indicator selection.

Second, the authors explained the literature review method and process and revised the first paragraph of section “2. Literature Review”. When conducting the literature review and selecting the evaluation index, the authors focused on the externalities of highway infrastructures not the externalities of general transportation infrastructures. The author knows that different types of transportation infrastructures have different externalities or have different intensities of externalities. For example, there is a big difference between highway infrastructures and underground rail transit (URT). For example, highways cause much greater exhaust emissions than URT. Highway construction will occupy the aboveground land area and cause great ecological damage. Therefore, many shelter forest will be established. However, because the URT is built underground, the impact of URT to the ecological system can be very small. Highways connect different regions and contribute greatly to external economic relations. However, URT’s contribution to the economy is more within a city. Actually, the authors have systematically compared and studied the externalities of different infrastructure systems.

Third, the selected indicators and available data focus on the highway infrastructures. The authors accepted another reviwer’s suggestion to separate the impact of highway infrastructures to the external economic relations and updated this revision in the revised manuscript. Except the education, medical care, and internal economic driving, all other indicators are widely accepted externalities of highway infrastructures. As for  the education, medical care, and internal economic driving, they are the outcomes of all infrastructures of a region and it is generally unrealistic to completely isolated the impacts of each kinds of infrastructures since they together contribute to the development of a society and the quality of life of human beings.

  • The reviewer commented that the indicators chosen are highly correlated which may cause the problem of double-counting. The choice of indicators must be well-studied and justified which is amply not presented. For example, mobility and accessibility are considered in one indicator even though they are different.

Response: As explained in Comment 2 and in the revised manuscript, the authors selected the indicators following three principles: (1) the selection of the indicators is systematic; (2) the indicators are widely accepted as the externalities of highway infrastructures; and (3) there are available statistical data for each indicator. This study used a comprehensive literature to study and determine the suitability of the selected indictors. To further check the applicability of the indicators, this study also conducted interviews with experts in the transportation field. All of the above performance aims to eliminate the double-counting problem. Each indicator in the finalized index system evaluates the externalities of highway infrastructures of different natures from different dimensions. Of course, this study pointed out that the selected indicators in this study are currently the most prevailing indicators. The authors admitted that this is a limitation of this study and updated it in the revised manuscript. In the future, new and significant indicators should be selected into the evaluation index system with the social development and the development of highway infrastructures.

As for the mobility and accessibility, they are not considered in one indicator. Generally speaking, the mobility and accessibility of highway infrastructures together contribute to the development of the economy and the society. The accessibility that significantly contributes to the positive social externalities of highway infrastructures was clearly explained in the original manuscript. As for the mobility, the authors much appreciated the reviewer for raising this question. The authors researched the knowledge of mobility. It is well accepted that mobility is often closely linked to one’s independence, well-being, and quality of life. In recent years, the transportation industry in Europe, America, Japan, and other countries has put forward the concept of mobility. For example, New York Government (2022) proposed the goal of efficient mobility in OneNYC2050 plan. The goal of efficient mobility includes congestion-mitigation, safety, health, and sustainability. If the mobility brought by highway infrastructures is efficient, it can bring many benefits such as travel convenience and time saving. However, due to the rapidly increasing in car ownership and the concerns of global warming, negative externalities brought by highway infrastructures, such as congetstion, accidents, air pollution, and noise, are widely criticized. Although these negative externalities were well explained in the original manuscript, the authors failed to use the word “mobility” to emphasize it. To clearly point out and emphasize the “mobility”, the authors revised the section 3 “(1) Evaluation indicators in the social dimension”. The revised paragraph is just below Table 1. In addition, after another round of literature review, the authors found that another important component of mobility that is travel costs should be added into the evaluation index system. This study proposed the unit traffic costs that is toll revenue divided by road length and number of civilian cars to represent the travel affordability of highway infrastructures. The authors also revised the section 3 “(2) Evaluation indicators in the economic dimension” in the revised manuscript.

Reviewer 2 Report

Dear Authors,

I appreciate the effort, work and idea. However, I have a few comments about which below.

The subject of the work is covered by research already undertaken in hundreds of articles / works (1) in the field of evaluation of the impact of transport with the inclusion of three axes with the addition of mathematical modeling. We have a social, environmental and economic perspective in many works (2). The evaluation of the impact on the needs of transport projects also has a rich literature for over 20 years (3).

(1) Zhu, L .; Ye, Q .; Yuan, J .; Hwang, B.-G .; Cheng, Y. A Scientometric Analysis and Overview of Research on Infrastructure Externalities. Buildings 2021, 11, 630. https://doi.org/10.3390/buildings11120630

(2) Chatziioannou, I .; Alvarez-Icaza, L .; Bakogiannis, E .; Kyriakidis, C .; Chias-Becerril, L. A Structural Analysis for the Categorization of the Negative Externalities of Transport and the Hierarchical Organization of Sustainable Mobility's Strategies. Sustainability 2020, 12, 6011. https://doi.org/10.3390/su12156011

(3) Shiftan, Y., Ben-Akiva, M., De Jong, G. et al. (2002) Evaluation of Externalities in Transport Projects. European Journal of Transport and Infrastructure Research, 2 (3/4). pp. 285-304.

Interesting in the aspect of creating another research model, the work gives us a tool that can help planners rather than politicians (line: 602).

In my opinion, correct in terms of the methods used. However, I agree with the authors' reservations in the context of limitations (line: 606-609), which basically undermines its applicability even in a three-axis approach. The selection of indicators for assessing the external effects of road infrastructure is debatable, but I respect the Authors' rights to the selection they consider important (Table 1, line: 429).

I do not understand why there is no economic dimension in negative externality - after all, we have losses in the tourism sector or a decrease in the value of land for housing development due to the course / construction of the highway infrastructure. Also, the economic factor calculated in GDP per capita (Table 1) does not show the reality. For China, it seems to me that it is better suited to the reality of PPP per capita. China (2021) GDP per capita: $ 12,550; PPP per capita: $ 17,700, that's too big a difference (40%) not to be included [based on https://tradingeconomics.com/china/gdp-per-capita-ppp].

As understood by “Total import and export of goods” (table 1) for the analyzed province, it covers only road transport, without rail or air. In fact, the use of one variable with a questionable value in the model to some extent weakens the model, but as I have already written, I recognize the authors' right to select variables.

It also seems to me that the title of the work could be better chosen. In the second paragraph of the work, the authors emphasize the model approach and emphasize its applicability, similarly to the abstract (points 1-3), which is not reflected in the title of the work, which is limited to a case study.

In terms of technical comments - literature to be checked and corrected. In my opinion, items 5, 16, 21 need improvement.

I also suggest extending the literature - at the authors' decision, perhaps the works of cf. Zhu, L .; Ye, Q .; Yuan, J .; Hwang, B.-G .; Cheng, Y. A Scientometric Analysis and Overview of Research on Infrastructure Externalities. Buildings 2021, 11, 630. https://doi.org/10.3390/buildings11120630

Best regards and, despite my remarks, I wish you success.

Author Response

Response to Reviewers

Dear Editors and Reviewers,

We would like to thank you for the positive and constructive comments and suggestions. We have substantially revised our manuscript according to the valuable comments. The point-to-point responses to the comments are listed as follows. The revisions made to the manuscript can be easily seen because the manuscript is using the “Track Changes” function. For references that were deleted and cannot be shown in the manuscript, the authors marked the corresponding places with “Yellow” color.

Response to Reviewer #2’s comments:

  • The reviewer commented that there are already hundreds of articles/works related to the impact of transport, the needs of transport projects, and the externalities of transport project. This study creates another research model and gives a tool that can help planners rather than politicians. For the externalities of transport project, the reviewer provided three relevant articles to the authors.

Response: First of all, thank the reviewer for providing us an opportunity to further explain the reason for creating another research model. Considering the reviewer’s questions and the recommended articles, the authors conducted another thorough literature review of the externalities of transport project and revised the section of “2. Literature Review” and the section of “3.1 Construction of Three-dimensional Evaluation Index System”. The authors also further explain the reason for creating this research model. The revision can be seen in the last paragraph of section “2. Literature Review” in the revised manuscript. From a macro and holistic perspective, the authors used the Social-Economic-Natural Complex Ecosystem theory to explain the needs for the three-dimensional evaluation. From a micro and technical perspective, due to the different forms of the externalities of highway infrastructures, there are few studies to quantify the externalities of highway infrastructures with unified dimensional indicators, decreasing the comparability of evaluation results. The ultimate purpose of the evaluation of the externalities of highway infrastructures is not only to reveal the impacts of all aspects of externalities but also to reveal how to balance the impacts of all aspects of externalities.

  • The reviewer agreed with the authors' reservations in the context of limitations (line: 606-609). However, the reviewer thought that it undermines its applicability even in a three-axis approach. The reviewer commented that the selection of indicators for assessing the external effects of road infrastructure is debatable. However, the reviewer respected the Authors' rights to the selection that they consider important (Table 1).

Response: Generally speaking, infrastructure is a system. Only when they work together, they can promote social development. However, through an in-depth literature review and three-year research project, the authors found that the externalities of various infrastructure are very different. It is difficult to form a model to evaluate externalities of different infrastructure system. Scholars in various field pay more attention to the externalities of the infrastructure that they are familiar with, but do not completely distinguish the contribution rate of various types of infrastructure.

In terms of the selection of the indicators, the authors revised the section of “3.1. Construction of Three-dimensional Evaluation Index System” to make it more systematic and realistic. Previously, the authors refereed to the theories of social welfare system to ensure the systematicness of the indicator selection which makes the reviewer feel less systematic and specific. It is widely acknowledged that the ultimate aim of highway infrastructure development that is to support social development and improve the quality of life of human beings. The authors conducted another round of literature review about the social indicators. The authors found that Human Development Index (HDI), Social Progress Index (SPI), Gallup-Healthways Well-Being Index, Canadian Index of Well-being (CIW), and the OECD’s better life index are well-influential social indicators. Comparing our developed index system and the above social indicators, we ensured that our index system is systematic. Our developed index system includes (1) the basic medical care, income and wealth, jobs and earnings, personal safety that represent the basic human needs, (2) the access to basic knowledge, health and wellness, and ecosystem sustainability that represent the foundations of wellbeing, and (3) the freedom of travel and the access to advanced education that represent the opportunity. The authors also adopted the theory of Social-Economic-Natural Complex Ecosystem that emphasizes the synergy of social, economic, and ecological systems when developing the index system.

Of course, this study pointed out that the selected indicators in this study are currently the most prevailing indicators. The authors admitted that this is a limitation of this study and updated it in the revised manuscript. In the future, new and significant indicators should be selected into the evaluation index system with the social development and the development of highway infrastructures.

  • The reviewer commented that why there is no economic dimension in negative externality - after all, we have losses in the tourism sector or a decrease in the value of land for housing development due to the course / construction of the highway infrastructure.

Response: The author’s question gives us good enlightenment. It let us rethink our research perspective. In fact, the comprehensive evaluation model is to evaluate the externalities of highway infrastructure from the perspective of a region which was emphasized in the first paragraph of section “3.1. Construction of Three-dimensional Evaluation Index System”. Taking a province for an example, in theory, improving highway infrastructure of a province will increase tourism opportunities and bring benefits to the tourism industry. Of course, this assumption should also be based on the good tourism resources in this area. Otherwise, the opposite is true. The development of tourism plays a leading role in the economy of a province. This study does not subdivide the contribution of highway infrastructure to various industries such as the tourism industry and the real estate industry because these are basically reflected in the economic development of a region. Therefore, this study reflects the impact of highway infrastructure on economy using two indicators that are external economic relations and internal economic driving.

As for the value of land that is used for the development of highway infrastructure, it is actually a very interesting topic. Our research team has conducted a review on this topic and summarized three aspects of research. First, the article recommended by the reviewer, which is “a scientometric analysis and overview of research on infrastructure externalities” written by Zhu, et al. (2021), clearly showed that there are numerous articles on the impact of transportation infrastructure on the house prices caused by the commuting convenience or accessibility. Researchers explored valuating such positive externalities through spatial econometrics and hedonic price model. In addition, such kind of research occupied the fourth largest cluster of the most influential documents in infrastructure externalities research. Second, the changes of the house price or the value of land are different at different stages. Researchers have explored the impacts of underground rail transit (URT) with respect to the spatial and temporal dimensions using the hedonic price model and panel data. The results not only proved the significant positive impact on the residential property values but also revealed that the influencing strength of the URT during its early stages of planning and construction was higher than that during the operation period (Ke and Gkritza, 2019; Zhang and Jiao, 2019). From this perspective, the increase in value is not a complete change in accessibility but a price change driven by speculation or investment. Of course, living closet to the transport infrastructure during the construction and operation period may have viewed as a nuisance than a benefit. Such kind of research are mainly from the perspective of local externalities caused by transportation infrastructure. It is worth noting that, if taking a city as a system, the increase of housing price in one area caused by the development of transportation infrastructure nearby may lead to the decrease of house price in other areas. From the perspective of a city, the improvement of transportation infrastructure will improve the image of the city and promote economic growth. The price of real estate is affected by too many factors, the two most important of which are economy and population. Considering the above, the authors believe that the economic improvement of the whole region should be taken as the key variable. To explain the above reason in detail, the authors further explain why the change of the value of land for housing development due to the course/construction of the highway infrastructure are not include in the evaluation index system in the revised manuscript in the section of 3.1 “(2) Evaluation indicators in the economic dimension”. Third, the development of highway infrastructure deprives other approach of land use such as for crops and real estate development (Mansuroglu, et al., 2013).  Spatial distortions in land settlement patterns are also negative externalities of transportation infrastructure (Guerra, 2011). However, there are relatively few studies in this field.

  • The reviewer commented that the economic factor calculated in GDP per capita (Table 1) does not show the reality. For China, it seems to me that it is better suited to the reality of PPP per capita. China (2021) GDP per capita: $ 12,550; PPP per capita: $ 17,700, that's too big a difference (40%) not to be included [based on https://tradingeconomics.com/china/gdp-per-capita-ppp].

Response: Thank the reviewer for this constructive suggestion. PPPs, which is purchasing power parities, are the rates of currency conversion that try to equalize the purchasing power of different currencies. The authors adopted the reviewer’s suggestion for two reasons. First, PPPs eliminate the differences in price levels between countries. Of course, prices for goods and services are driven upward or downward because of numerous factors at work within the economy. However, we can, at least, use PPPs or inflation rate to eliminate the differences in price levels, reflecting the true increase of people’s purchasing power. Second, this study used ten thousand dollars per capita as the unit of another economic indicator which is external economic relations. We also intend to adopt PPPs in order to make the two economic indicators consistent. Therefore, this study adjusted the GDP per capita with PPPs using the PPP conversion rates based on OECD Data [https://data.oecd.org/conversion/purchasing-power-parities-ppp.htm].

  • The reviewer commented that the indicator “total import and export of goods” (table 1) for the analyzed province covers only road transport, without rail or air. The use of one variable with a questionable value in the model to some extent weakens the model.

Response: The authors are much appreciated for the reviewer’s question that provides us a good perspective to improve the indicator. Previously, the data of the total import and export of goods covers all modes of transportation. The authors did not make any modification of this data because we used the entropy weight method that is sensitive to the dispersion of the index data. There is an assumption that the contributions of different modes of transportation in a region relatively won’t change much. If multiplying the current data with a constant factor representing the contribution of road transport in a region, the relative weight of this indicator will have no difference with that using the current data. According to the reviewer’s emphasis and reminder, the authors searched and analyzed relevant statistical data which are freight volume (unit: ten thousand tons) and cargo turnover (unit: 100-million-ton kilometer) for different transportation modes. Comparing these two variables, cargo turnover is believed more accurate because it considers not only weight but also distance. After analyzing the data, the authors found that the contribution of different modes of transportation varied significantly in different years. We adjusted the indicator by multiplying the proportion of goods turnover of highway in the total turnover of various transportation modes to accurately represent the contribution of highway infrastructures to external economic relations. The adjustment makes the evaluation more realistic. This change was also explained in the revised manuscript in the section of 3.1 “(2) Evaluation indicators in the economic dimension”.

  • The reviewer suggested that the authors chose a better title of the work. In the second paragraph of the work, the authors emphasized the model approach and emphasize its applicability, similarly to the abstract (points 1-3), which is not reflected in the title of the work, which is limited to a case study.

Response: The authors much appreciated the reviewer for this constructive suggestion. The previous title decreased the value of this research. The new title was changed to “A Three-dimensional Evaluation Model of the Externalities of Highway Infrastructure to Capture the Temporal and Spatial Distance to Optimal”.

  • The reviewer suggested the authors to check and correct the literature, such as items 5, 16, 21.

Response: The authors made a thorough check of the literature and corrected all errors.

  • The reviewer suggested that the authors extend the literature.

Response: Thanks for the reviewer’s suggestion. As the response to Comment 2, the authors actually conducted another round of literature review.

References:

 Guerra, E. (2011). "Valuing rail transit: comparing capital and operating costs with consumer benefits." Transp. Res. Record, 2219(1), 50-58.

 Ke, Y., and Gkritza, K. (2019). "Light rail transit and housing markets in Charlotte-Mecklenburg County, North Carolina: Announcement and operations effects using quasi-experimental methods." J. Transp. Geogr., 76, 212-220.

 Mansuroglu, S.,Kinikli, P., and Yilmaz, R. (2013). "Impacts of highways on land uses: the case of Antalya-Alanya highway." Journal of Environmental Protection and Ecology, 14(1), 293-302.

 Zhang, D., and Jiao, J. (2019). "How does urban rail transit influence residential property values? Evidence from an emerging Chinese megacity." Sustainability, 11(2), 534.

 Zhu, L.,Ye, Q.,Yuan, J.,Hwang, B.-G., and Cheng, Y. (2021). "A Scientometric Analysis and Overview of Research on Infrastructure Externalities." Buildings, 11(12), 630.

Reviewer 3 Report

Dear Author(s), I would like to thank you for the opportunity to read your manuscript entitled “Three-dimensional Evaluation of Externalities of Highway Infrastructures - A Case Study of Jiangsu Province”.

The overall manuscript is well presented with minor spelling or grammar mistakes, especially those with an inappropriate article used.

The overall work is very interesting, as the problem of evaluation of externalities of highway infrastructures, hindering the sustainable planning and development of highway infrastructure is very relevant.  

Here are some issues concerning your paper:

  1. The overall purpose of the article is stated clearly in the introduction and underlined in the abstract.
  2. The Literature Review part is logical and well organized. There is a lot of research into road safety and infrastructure solutions that can improve it. Wouldn't that be worthwhile in terms of social or economic factors where external costs are considered? Perhaps it would also be worth quoting, for example: Cieśla, M. Modern Urban Transport Infrastructure Solutions to Improve the Safety of Children as Pedestrians and Cyclists. Infrastructures 2021, 6, 102. https://doi.org/10.3390/infrastructures6070102
  3. All Figures are well explained and done with particular quality.
  4. Future research directions and the significance of the results of the research achieved should be underlined and explained in conclusion part.

Reviewer

Author Response

Response to Reviewers

Dear Editors and Reviewers,

We would like to thank you for the positive and constructive comments and suggestions. We have substantially revised our manuscript according to the valuable comments. The point-to-point responses to the comments are listed as follows. The revisions made to the manuscript can be easily seen because the manuscript is using the “Track Changes” function. For references that were deleted and cannot be shown in the manuscript, the authors marked the corresponding places with “Yellow” color.

Response to Reviewer #3’s comments:

  • The reviewer commented that the overall work is very interesting and the overall manuscript is well presented. However, there are some minor spelling or grammar mistakes, especially those with an inappropriate article used.

Response: First of all, thank you very much for your affirmation of this study, strengthening the resesarchers’ confidence in continuing relevant research work. For the spelling and grammar mistakes, the authors carefully and thoroughly checked the manuscript and correct all spelling and grammar mistakes.

  • The reviewer commented that the overall purpose of the article is stated clearly in the introduction and underlined in the abstract. Moreover, the Literature Review part is logical and well organized. However, the reviewer suggested that there is a lot of research into road safety and infrastructure solutions that can improve it. The reviewer recommended that the authors consider road safety where external costs are considered because it is worthwhile in terms of social or economic factors? Perhaps it would also be worth quoting, for example: CieÅ›la, M. Modern Urban Transport Infrastructure Solutions to Improve the Safety of Children as Pedestrians and Cyclists. Infrastructures 2021, 6, 102. https://doi.org/10.3390/infrastructures6070102.

Response: Thank you for the constructive comments and suggestions. Road safety is an integral part of the highway infrastructure. If improperly managed, as a negative externality of highway infrastructure, traffic accidents may occur. By adopting the reviewer’s suggestion, this study emphasized this in the section of “3.1. Construction of Three-dimensional Evaluation Index System”  in the paragraph just below Table 1. Moreover, this study quoted more relevant literature including  CieÅ›la, M.’s research.

  • The reviewer commented that all Figures are well explained and done with particular quality. However, future research directions and the significance of the results of the research achieved should be underlined and explained in conclusion part.

Response: Thank you for the constructive suggestions. By adopting the reviwer’s suggestion, this study explained the future research directions at the end of this manuscript which is in the conclusion part.

Round 2

Reviewer 1 Report

I want to commend you on addressing the comments properly. The revised manuscript is presenting a well-structured research that has a lot of potentials.   

Author Response

Comment 1: I want to commend you on addressing the comments properly. The revised manuscript is presenting a well-structured research that has a lot of potentials.

Response: We are very glad to get your recognition. We really appreciate the positive and constructive comments given by you and the other reviwers, giving us the opportunity to improve the quality of our article.